# Influence of Gestational Age on Pelvic Floor Muscle Activity, Plantar Contact, and Functional Mobility in High-Risk Pregnant Women: A Cross-Sectional Study

**DOI:** 10.3390/s24144615

**Published:** 2024-07-17

**Authors:** Emilly Cássia Soares Furtado, Yury Souza De Azevedo, Deizyane dos Reis Galhardo, Iasmin Pereira Cabral Miranda, Maria Eunice Chagas Oliveira, Pablo Fabiano Moura das Neves, Lindinalva Brasil Monte, Erica Feio Carneiro Nunes, Elizabeth Alves Gonçalves Ferreira, Bianca Callegari, Givago da Silva Souza, João Simão de Melo-Neto

**Affiliations:** 1Institute of Health Sciences, Federal University of Pará (UFPA), Belém 66075110, PA, Brazil; emillycsoares@outlook.com (E.C.S.F.); yurysouza0701@gmail.com (Y.S.D.A.); deizyanegalhardo@gmail.com (D.d.R.G.); pcabralmirandaiasmin@gmail.com (I.P.C.M.); mariaoliveira.fisio@gmail.com (M.E.C.O.); callegari@ufpa.br (B.C.); givagosouza@gmail.com (G.d.S.S.); 2Santa Casa de Misericórdia Foundation of Pará, Belém 66055080, PA, Brazil; pablo.neves@santacasa.pa.gov.br (P.F.M.d.N.); lindibrasil@gmail.com (L.B.M.); 3Department of Human Movement Sciences, State University of Pará, Belém 66087662, PA, Brazil; ericacarneiro@uepa.br; 4Faculty of Medicine FMUSP, University of São Paulo (USP), São Paulo 01246904, SP, Brazil; elferreira@usp.br

**Keywords:** high-risk pregnancy, locomotion, foot, pelvic floor

## Abstract

During pregnancy, biomechanical changes are observed due to hormonal and physical modifications, which can lead to alterations in the curvature of the spine, balance, gait patterns, and functionality of the pelvic floor muscles. This study aimed to investigate the progressive impact of biomechanical changes that occur during gestational weeks on the myoelectric activity of the pelvic floor muscles, plantar contact area, and functional mobility of high-risk pregnant women. Methods: This was a cross-sectional observational study carried out from November 2022 to March 2023. A total of 62 pregnant women of different gestational ages with high-risk pregnancies were analyzed using surface electromyography to assess the functionality of the pelvic floor muscles, plantigraphy (Staheli index and plantar contact area), and an accelerometer and gyroscope using the timed up and go test via an inertial sensor on a smartphone. Descriptive statistics and multivariate linear regression analyses were carried out to test the predictive value of the signature. Results: Increasing weeks of gestation resulted in a decrease in the RMS value (β = −0.306; t = −2.284; *p* = 0.026) according to the surface electromyography analyses. However, there was no association with plantar contact (F (4.50) = 0.697; *p* = 0.598; R^2^ = 0.53). With regard to functional mobility, increasing weeks of gestation resulted in a decrease in time to standing (β = −0.613; t = −2.495; *p* = 0.016), time to go (β = −0.513; t = −2.264; *p* = 0.028), and first gyrus peak (β = −0.290; t = −2.168; *p* = 0.035). However, there was an increase in the time to come back (β = 0.453; t = 2.321; *p* = 0.025) as the number of gestational weeks increased. Conclusions: Increased gestational age is associated with a reduction in pelvic floor myoelectric activity. The plantar contact area did not change over the weeks. Advancing gestation was accompanied by a decrease in time to standing, time to go, and first gyrus peak, as well as an increase in time to come back.

## 1. Introduction

During pregnancy, biomechanical changes are observed as a result of hormonal and physical modifications that can generate changes in the curvature of the spine and the balance and gait patterns of women [1,2]. Some pregnant women are prone to developing a high-risk pregnancy during the gestational period, which is characterized by medical or obstetric conditions that pose a real or potential danger to the health and well-being of the mother and fetus, requiring specialized attention and personalized care [3]. Among the factors that contribute to this risk are chronic health conditions such as diabetes and hypertension, obesity, multiple pregnancies, and very young or advanced maternal age [4].

In addition, kinematics can show changes in hip, knee, and ankle movements in all planes, along with variations in trunk kinematics associated with interactions with the pelvis [1]. These changes can impact the quality of life of pregnant women in various ways, resulting in increased reports of pain, difficulty maintaining suitable ergonomic conditions, and an increased risk of falls [5].

The function of the human foot is influenced by its anatomical structure according to the shape of the plantar arch and the points of support in the heel and metatarsal region that allow the weight of the body to be supported without foot collapse [6]. Many techniques are used to analyze the plantar footprint, such as plantigraphy, which provides the characteristics of the feet analyzed through impressions. In addition, the Staheli index is used, which evaluates the plantar arch through the relationship between the width of the arch region and the width of the heel region obtained from the plantar impression to determine whether the foot is cavus, flat, or normal [7].

To improve stability, pregnant women tend to modify their foot support and load distribution between the hindfoot, midfoot, and forefoot, as well as making their steps shorter and with greater ground contact time [8]. As a result, the gait pattern becomes lateralized, with an increase in peak time in the lateral midfoot region and an increase in the plantar area of up to 10%, which may be related to the strategies developed during pregnancy with weight gain [9]. Gijon-Nogueron et al. [10] noted that between the 12th and 34th weeks of pregnancy, pregnant women’s feet are prone to flattening and adopting a more pronated posture. It can also be seen that increased abdominal circumference impacts the risk of flattening of the medial arch, resulting in unstable gait [11]. Another factor that impacts the gait of pregnant women is posteriorization of the head and trunk, which is an adaptation for balance, with increased lateral displacement and lower limbs with an enlarged base, resulting in more movement in the frontal plane [2].

Maintaining balance during walking is fundamental for maintaining the integrity of the lower limbs and spine, which support ground motions [12]. To ensure this stability, adequate control of the pelvic region, which is responsible for efficiently distributing the weight of the trunk and upper limbs to the lower limbs, is essential [13]. Although the trunk muscles play a crucial role in support, poor control of the pelvis can lead to overload and structural dysfunction. Pelvic stability occurs due to the synergy between the passive (bones, joints, and ligaments), active (muscles and fascia), and control (neural system) systems, which act in coordination to maintain the harmony and efficiency of movement [14].

Functional tests are carried out to assess the physical fitness and mobility of pregnant women. The timed up and go (TUG) test starts with the patient sitting with her arms on the armrests and her toes against the starting line and aiming to walk for approximately 3 m, turn around, and return to the chair [15]. Evensen NM, Kvåle A, and Brækken IH in Ref. [16] showed that the TUG test is reliable for pregnant women. In addition, pregnant women walk more slowly and may be unable to complete the TUG test [17]. Smartphone inertial sensors are currently used as technological support to help measure an individual’s speed or calculate the angles of movement during a mobility test [18].

The changes that occur during pregnancy affect the performance of functional activities, such as moving from sitting to standing [19]. In this context, the ability to get up from a sitting position requires postural and motor control, and this control usually becomes deficient as pregnancy progresses [20]. A sit-to-stand movement is defined as the transfer of the body’s center of mass from a seated position, where there is a large base of support, to a stable standing position, with the achievement of orthostatic balance and the ability to control body sway [21]. During pregnancy, difficulty executing movements is also influenced by other physiological factors, such as decreased proprioception, muscle strength, postural balance deficit, and joint pain, thus limiting the pregnant woman’s range of movement [21].

During pregnancy, there is a physiological increase in ligament laxity [22] due to hormonal action, which is associated with lumbar hyperlordosis and an enlarged abdomen, leading to a reduction in the effectiveness of the pregnant woman’s passive stability system, which must be compensated for by dynamic stability through increased activity of the local muscles [23]. Among these muscles are the pelvic floor muscles (PFMs), which can be dysfunctional due to physiological changes and are important lumbar–pelvic stabilizers. When the structure of the pelvic floor is altered, dysfunctions such as urinary incontinence, fecal incontinence, prolapse of pelvic organs (uterus or other pelvic organs), and chronic pelvic pain may occur [24].

Surface electromyography (sEMG) is appreciated as a tool for real-time assessment of PFM contractions and for functional analysis of this area because it identifies the action potential of PFM motor units [25]. The electrical signals of muscles are produced from the recruitment of motor units when contraction occurs; the bioelectrical activity of a muscle can be observed as a representation of muscle function [26]. A study by Resende et al. [27] compared pregnant women and nulliparous women via electromyography and vaginal palpation of PFMs and showed that pregnant women have worse PFM function with decreased strength and electrical activity and that these changes can persist during pregnancy or persist throughout a woman’s life.

No studies have evaluated these variables according to gestational age, the myoelectric activity of the pelvic floor muscles, the plantar contact area, or functional mobility in high-risk pregnant women. Therefore, this study aimed to investigate the progressive impact of biomechanical changes that develop during gestational weeks on the myoelectric activity of the pelvic floor muscles, the plantar contact area, and functional mobility in high-risk pregnant women. Our initial hypothesis was that there would be an increase in the EMG of the pelvic floor muscles to increase the lower support of the pregnant uterus, an increase in plantar contact area and plantar arch collapse to increase the support base, and that there would be a reduction in all parameters of functional mobility during the TUG test.

## 2. Materials and Methods

### 2.1. Ethical Aspects

This study was conducted after approval by the Ethics and Research Committee of the Fundação Santa Casa de Misericórdia do Pará under protocol no. 5.081.508 and the signing of the informed consent form (ICF). The individuals were studied in accordance with the Declaration of Helsinki and the norms of research involving human beings (Res. 466/12 and Res. 510/16 of the National Health Council).

### 2.2. Study Design

This was a cross-sectional observational study of the analytical, descriptive, and inferential type. The study was carried out in accordance with The Strengthening the Reporting of Observational Studies in Epidemiology (STROBE) Statement: guidelines for reporting observational studies.

### 2.3. Setting and Period of Study

The collections were carried out in Fundação Santa Casa de Misericórdia do Pará, a public maternal and child hospital in Belém, PA. The study period was from November 2022 to March 2023.

### 2.4. Population

The participants were women with high-risk pregnancies aged 18 years or older.

### 2.5. Sampling

The convenience type of sampling is nonprobabilistic.

### 2.6. Sample Size

For the sample size calculation, the parameters related to the model for predicting the myoelectric activities of the pelvic floor muscles (partial R^2^ = 0.361) were selected according to the study developed by Duarte et al. [28]. The calculated effect size f^2^ was 0.565. The established parameters were two-tailed analyses, with a probability of error of α = 0.05 and β = 0.2 for 5 predictions. A total minimum sample size of 17 participants was obtained. The sample size was calculated using G* Power software version 3.1.9.7.

### 2.7. Eligibility Criteria

This study included women with a medical diagnosis of high-risk pregnancy aged ≥18 years who agreed to participate in the study by signing an informed consent form. Pregnant women under 18 years of age were excluded because of a lack of contractility of the pelvic floor muscles, neurological dysfunctions, urinary or vaginal infection or any other pelvic condition that may alter electromyography results, mobility difficulties not related to pregnancy, incomplete physical assessment, or refusal to participate in the study. The sample eligibility flowchart is shown in Figure 1.

### 2.8. Instruments and Variables

#### 2.8.1. Socioeconomic Form

To characterize the socioeconomic status of the patients, information was collected using a form developed by the authors. The variables collected were age (years), race/skin color, marital status, education (years), and family income (Brazilian minimum wage of R$ 1212.00).

#### 2.8.2. Surface Electromyography

Electromyography (EMG) is the most objective and reliable method available for obtaining information on muscle function and efficiency and is performed by identifying muscle electrical potentials [29]. This EMG stage is shown in Figure 2. Figure 2A provides an overview of the implementation.

The EMG System do Brasil^®^ (EMG (EMG SYSTEM DO BRASIL Ltd., São Paulo, Brazil)) was used to collect the data, characterized by a 4-channel analog–digital converter with 16-bit resolution and an input range of −12 to +12 volts, a sampling rate of 2 kHz and a frequency in the 20 to 500 Hz range. For detailed analysis of the myoelectric signals, as shown in Figure 2B, a computer was connected to the device, which displayed the data at a rate of 2000 Hz. In addition, MedPex^®^ (Electrode (MedPex by DBIMedical Ltd., São Paulo, Brazil)), a self-adhesive electrode with a circular shape of 40 mm composed of Ag/AgCl and a solid hydrogel adhesive, was used. This adhesive is composed of carboxymethylcellulose, glycol, and preservatives. For the preliminary analysis described in this study, the raw data provided by the device were used.

The sEMG recording methodology, including electrode placement, EMG signal processing, and modeling, is based on the recommendations of Surface Electromyography for Non-Invasive Assessment of Muscles (SENIAM) and the International Society of Electromyography and Kinesiology (ISEK). However, there is still no methodological standard or consensus for positioning electrodes in the pelvic floor region [30]. Therefore, in this study, the active electrodes were placed on the muscles of the perineal region, as shown in Figure 2B, at a distance of 20 mm, and the reference electrode was located on the right fibular malleolus, as performed in our previous study [31].

During the assessment, the pregnant women were instructed to lie down in a supine position, with their hips and knees flexed to approximately 90° [32] (Figure 2A). For the EMG recordings, the patients were instructed to perform three maximum voluntary contractions, followed by relaxation, with 10 s of rest between contractions. At the end, the best of the three contractions was selected for the study [33].

The raw electromyographic signals were filtered in the 20 and 400 Hz range, full-wave rectified, and bidirectionally filtered with a 100 Hz low-pass Butterworth filter without delay. The root mean square (RMS) value was calculated using a 250-point window. After this step, a second-order low-pass Butterworth filter with a zero delay of 6 Hz was used to smooth the signal, and the EMG signal was integrated during the same contraction interval (0.5–4.5 s) (Figure 2C).

For the analysis of electromyographic signals, the following parameters were extracted: the RMS of the 5 s contraction period, expressed in microvolts (μV); peak RMS values, also expressed in μV; area values, also in μV; the percentage of maximum voluntary contraction (MVC), normalized by the signal peak; and the median frequency of the signal, obtained after applying the fast Fourier transform.

#### 2.8.3. Plantigraphy

The function of plantigraphy is to take a paper impression of the plantar surfaces of the feet under the load of body weight [34]. It is a technique for analyzing the footstep that allows the observation of characteristics such as the type of arch, the distribution of plantar pressure, the inclination of the heels, the position of the metatarsals, and other relevant information [35]. The procedure is shown in Figure 3.

The device (Figure 3A) has two rectangular boards that sit in the middle of a rubber sheet structured on the inside into squares filled with smaller squares. This is where the water-soluble ink is applied and where the paper is placed to make the plantar impression (Figure 3C). To obtain an accurate footprint, the pregnant woman was initially seated, the podograph was placed under the foot, and then the person stood, allowing the weight of the body to fall on the platform (Figure 3B) [6].

The images were scanned and saved in a .jpg file. The study of the plantar footprints was carried out through two measurements:The Staheli index was used to carry out the analysis because it was obtained by dividing the narrowest part of the isthmus by the value of a parallel at the widest part of the heel. The reference values used for evaluation are in the normal range of 0.6 to 0.69, while values greater than 0.69 indicate cavus feet, and values less than 0.6 indicate flat feet [6] (Figure 3D).The total area of both sides of the feet was analyzed using ImageJ software, version 1.47 (National Institutes of Health, Bethesda, Maryland) (for 32-bit windows), with image sizes of 531,529 bytes [36] (Figure 3E).

#### 2.8.4. Inertial Sensors and Timed Up and Go Test

The TUG test was used to perform a movement analysis and identify different patterns related to angular velocity during gait (Figure 4A). The TUG test comprises five phases of execution: (1) the individual sits on a chair; (2) the individual stands up and walks 3 m; (3) the individual reverses his or her gait; (4) the individual returns; and (5) the individual sits back down on the chair. Throughout this test, movements can be measured with the help of inertial sensors that are available on smartphones [18]. This study used a mobile device (Android A10s, Samsung, Seoul, Republic of Korea) with accelerometry and gyroscopy sensors and octa-core processors with speeds of 2 GHz and 1.5 GHz. The footpad was taken at a sampling rate of 120 Hz. An application called “Momentum Science”, installed on a mobile device with an Android platform in the Java language, is responsible for storing the readings of the inertial sensors embedded in the smartphone [37].

The smartphone was placed on the back of the participant’s body, at the level of the lower lumbar spine, between the L3 and L5 vertebrae, and fixed to the body using a strong elastic band attached to the waist (Figure 4C). The data were transferred to the GNU OCTAVE program, where the accelerometer and gyroscope data were analyzed, as shown in Figure 4B.

### 2.9. Data Processing

The Momentum Science application exported the accelerometer and gyroscope records as text records, which were then exported and analyzed using routines in MATLAB (MATLAB R2015a, Mathworks, CA, USA). A linear trend extraction procedure was carried out on the inertial time series using the detrending function. The signals from the sensor were normalized by dividing them by 9.81 to express the data in gravitational units. The norm of the accelerometer and gyroscope signal vectors was then calculated using Equation (1): norm = √x^2^ + y^2^ + z^2^. The letters represent the accelerations and angular velocities in the lateral (x), vertical (y), and anteroposterior (z) axes. Linear interpolation of the resulting vectors was performed to select the data at an amplitude of 100 Hz. Nevertheless, we used a second-order bidirectional Butterworth filter with a cutoff frequency of 5 Hz to reduce possible noise and prevent aliasing.

During the test, 6 transient events were evaluated, as shown in Figure 4. Event 1 was the detection of the start of the TUG test by changing posture from sitting to standing. Event 2 was the act of walking three meters. Event 3 is the moment of turning back toward the chair. Event 4 involved walking three meters to sit down. Event 5 was the postural transition from standing to sitting. Event 6 refers to the end of the test.

The variables of interest were calculated based on the following identified temporal markers: test duration (s), time to stand (s), time to go (s), time to come back (s), first gyrus peak (g), standing jerks (g/s), and speed to go (rad/s). The analysis was carried out automatically by the standardized analysis routine and monitored by the researchers.

#### 2.9.1. Bias

The study design made it susceptible to sampling bias, which was minimized through the collection of data from only high-risk pregnant women. In addition, this study is also susceptible to collector bias, and to minimize this possibility, the collection was standardized, and the professionals trained, keeping the same collection team with each individual in their respective role.

#### 2.9.2. Statistical Analysis

Descriptive statistics were analyzed to determine the absolute and relative frequencies, means, and standard deviations (for parametric data) or medians and interquartile ranges (IQRs) (for nonparametric data) for each age group. Multivariate linear regression analysis was performed to assess the predictive value of the model.

To ensure the reliability of the model, the following acceptable criteria were established: the Durbin–Watson within-interval test (1.5;2.5) to demonstrate the independence of the residuals; Cook’s distance (less than 1) to confirm the absence of outliers in the data set that could jeopardize the estimation of the coefficients; and the value of the variance inflation factor (VIF) (less than 10) and tolerance (greater than 0.2) to verify the absence of multicollinearity, a Gaussian distribution (Figure 2A), and a P-P plot (Figure 2B) in which the comparison between “observed probability” and “expected probability” was used to test the normality of the residuals.

All the statistical analyses were performed using the Statistical Package for Social Sciences (SPSS for Windows, v21.0; IBM, Chicago, IL, USA) software, with the α value set at 0.05.

## 3. Results

The characteristics of the women evaluated are presented in Table 1. The mean age of the participants was 23 ± 8.5 years. The most frequent clinical diagnoses were gestational diabetes (n = 7; 11.3%), hypertension (n = 5; 8.1%), diabetes mellitus (n = 4; 6.5%), and uterus bicornuate (n = 3; 4.8%). The other clinical diagnoses were presented in fewer than three patients.

### 3.1. Myoelectric Activity of the Pelvic Floor Muscles

Initially, the residual assumptions were tested (Figure 5). The peak and area were excluded from the model due to multicollinearity with the RMS (VIF > 10; tolerance > 0.1). Subsequently, the analysis resulted in a statistically significant model (F (2.54) = 3.375; *p* = 0.042; R^2^ = 0.111), including only the RMS and MVC variables. An increase in gestation time resulted in a decrease in the RMS value (β = −0.306; t = −2.284; *p* = 0.026).

### 3.2. Plantar Contact

There was no association with plantar contact (F (4.50) = 0.697; *p* = 0.598; R^2^ = 0.53).

### 3.3. Functional Mobility

Initially, the residual assumptions were tested (Figure 6). The test duration was excluded from the model due to multicollinearity with the RMS (VIF > 10; tolerance > 0.1). Subsequently, the analysis resulted in a statistically significant model (F (6.46) = 2.670; *p* = 0.026; R^2^ = 0.258), including only the time to stand, time to go, time to come back, first gyrus peak, standing jerks, and speed to go variables. An increase in gestation time resulted in a decrease in the time to standing (β = −0.613; t = −2.495; *p* = 0.016), time to go (β = −0.513; t = −2.264; *p* = 0.028), and first gyrus peak (β = −0.290; t = −2.168; *p* = 0.035) values. However, there was an increase in the time to come back (β = 0.453; t = 2.321; *p* = 0.025) as the number of gestational weeks increased.

Figure 7 represents the results for the variables of electromyography, plantar contact and functional mobility.

## 4. Discussion

In this study, we examined the influence of the progression of physiological biomechanical changes that develop during gestational weeks on the bioelectrical activity of the pelvic floor muscles, plantar contact, and functional mobility in high-risk pregnant women. We found that an increase in gestational age was associated with a reduction in the myoelectric activity of the pelvic floor muscles, a decrease in “time to standing”, “time to go”, and “first gyrus peak” and an increase in “time to come back”.

During pregnancy, the mechanical properties of the pelvic floor adjust to facilitate vaginal delivery while protecting against birth injuries [38]. Routzong et al. [39] reported that pregnant women in the third trimester show alterations related to soft tissue remodeling influenced by hormonal and biochemical changes, which alter the mechanical properties of tissues, increasing the mobility of joints and organs such as the bladder/urethra and the distensibility of soft tissues. These factors, as pregnancy-specific changes, begin immediately in the first trimester and present changes related to the pelvic floor, such as bladder neck mobility [40] and reduced basal tone [41]. These tissue changes, mainly muscular changes, have been evaluated to understand the motor behavior of PFMs during pregnancy [42,43,44].

In our study, with regard to the electromyographic activity of the PFMs, we found that increasing gestational age resulted in a decrease in the RMS. Considering that the RMS is a factor for sEMG amplitudes and is used to assess the pattern of muscle activation, the recruitment of motor units and the firing rate are two main components of force generation, as reflected in the sEMG signals [45,46]. Prudencio et al. [42] also analyzed PFM activation patterns using electromyography signals and reported that pregnant women with diabetes mellitus had lower PFM activation levels than control pregnant women in the last weeks of pregnancy. It is possible that the relationship found in our study is because there is less activation of the muscle fibers, decreasing the functionality of the PFMs as gestational age advances. These results could contribute to improving clinical practice due to the characteristics of high-risk pregnancies and could help in screening for pelvic muscle dysfunction in pregnant women. Therefore, it is of fundamental importance that professionals work to rehabilitate these muscles and not just leave the patient to rest or undergo conservative treatment. Therefore, these clinical cases should be discussed with a multiprofessional team.

This study assessed the area of the plantar impression, followed by analysis of the “staheli” index measurement in high-risk pregnant women; however, there was no significant association related to plantar contact. Other studies have reported results different from ours using other assessment tools. The literature shows a significant decrease in the reduction of the longitudinal arch of the foot during the first trimester of pregnancy without high risk [13]. In addition, another study revealed that the height of the arch modifies the pattern of pressure distribution on the plantar surface [6]. Gaymer et al. [47] noted that midfoot plantar pressure increased significantly during late pregnancy. Such studies make it possible to understand biomechanical changes and how plantar contact changes. Although our study did not detect significant differences when investigating the progressive impact of the biomechanical changes of pregnancy on the plantar contact area with the tools used, it is important that other studies investigate the relationship in longitudinal studies, considering intragroup comparisons, to obtain results that strengthen clinical practice.

Previous studies have shown that gait kinematics also change during pregnancy. These changes include modifications in pelvic tilt, increased hip flexion, and increased range of motion of the knee in support and ankle sway in the sagittal plane [48]. These changes may be related to significant increases in weight and abdominal circumference, shifts in the center of gravity, increased body sway, and altered perception of balance [1,12]. Cakmak et al. [49] reported that postural stability is impaired due to decreased postural stability during pregnancy, particularly in the third trimester, which increases the risk of falling.

To assess postural changes, smartphone inertial sensors are routinely used in the results or monitoring of the patient during movement and rehabilitation, and the literature reports high reliability in the measurements obtained by smartphone gyroscopes and accelerometers as sensors to make a more accurate postural analysis [50]. There is a gap in the research on functional mobility using the TUG test in conjunction with a smartphone inertial sensor in pregnant women. In our study, it was possible to evaluate these methodological characteristics, and we found that an increase in gestational age resulted in a decrease in “time to standing”, “time to go”, and “first gyrus peak” during the TUG test. However, there was an increase in “time to come back” as gestation progressed.

These findings are similar to those of the studies by Tomoe Inoue-Hirakawa et al. [51], who analyzed the gait function of pregnant women and found that TUG performance was significantly slower in this population, which is related to mass gain during gestation. Our results suggest that the progression of pregnancy can affect different aspects of a pregnant woman’s functionality, demonstrating that postural adaptation can influence individual safety and mobility. For clinical practice, the use of inertial sensors in smartphones combined with the TUG could be an assessment tool to monitor these changes throughout pregnancy. In relation to “time to come back”, we believe that this is a way of compensating pregnant women to reduce the time it takes to perform the test.

It is important to recognize that this study has limitations because, although it involved high-risk pregnant women, it did not carry out a comorbidity analysis to determine whether the data on myoelectric activity, plantar contact, and functional mobility differed. It is suggested that other studies could verify this relationship for different comorbidities related to high-risk pregnancies. It is therefore important that future studies determine an interval for comparing these changes according to gestational week. Despite these limitations, we believe that this study contributes to understanding the biomechanical impacts on functionality and provides new evaluation approaches for high-risk pregnant women. Finally, longitudinal observational studies analyzing these variables could provide new perspectives on biomechanics.

## 5. Conclusions

We concluded that an increase in gestational age is associated with a reduction in pelvic floor myoelectric activity. No significant changes were observed in the plantar contact area of the high-risk pregnant women. Advancing gestation was accompanied by a decrease in the time to get up, time to go, and peak of the first turn, as well as an increase in the return time.

## Figures and Tables

**Figure 1 sensors-24-04615-f001:**
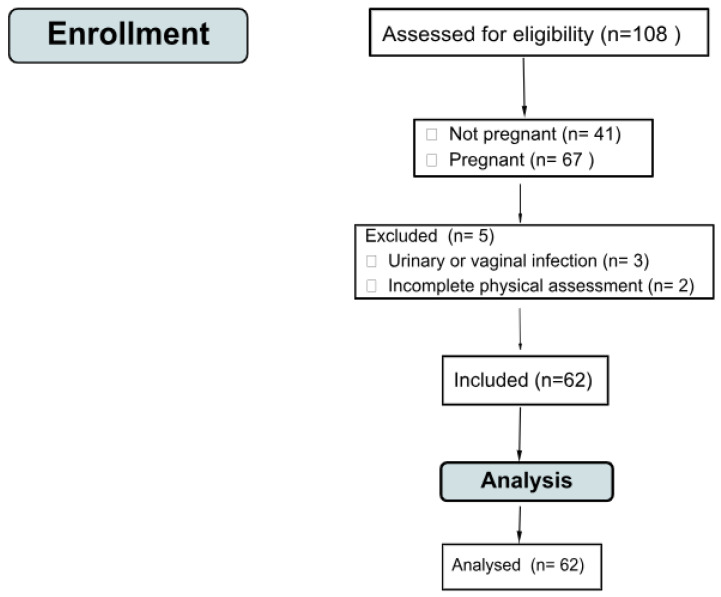
Eligibility flow diagram.

**Figure 2 sensors-24-04615-f002:**
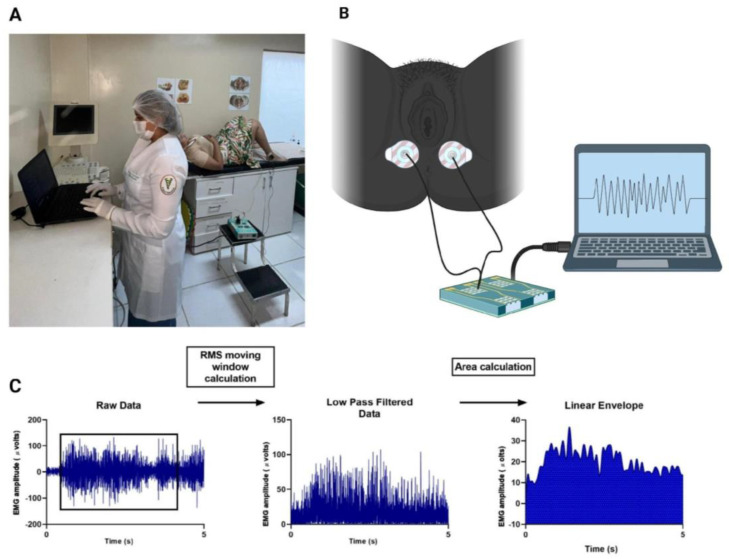
Demonstration of the collection of the EMG; (**A**) illustration of the data processed on the computer (**B**) and graphical representation of the treatment of the EMG signal. Created by BioRender.com. (**C**) Graphical representation of the EMG signal smoothed using a low-pass Butterworth filter.

**Figure 3 sensors-24-04615-f003:**
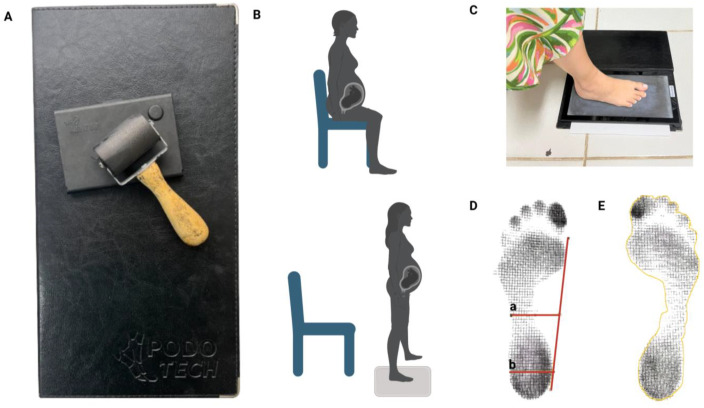
Summary of the plantigraphy procedure performed on the pregnant woman and the results of her foot impressions. PODOTECH plantigraphy equipment (**A**). Illustration of the execution of the plantar contact area (**B**). Representation of the collection of the plantar contact area in a volunteer (**C**). Analysis of the Staheli Index (**D**): measurement of the width of the central region of the foot (a), and of the heel Region (b), and of the plantar contact area in ImageJ (**E**). This tool was created by “BioRender.com” (2024).

**Figure 4 sensors-24-04615-f004:**
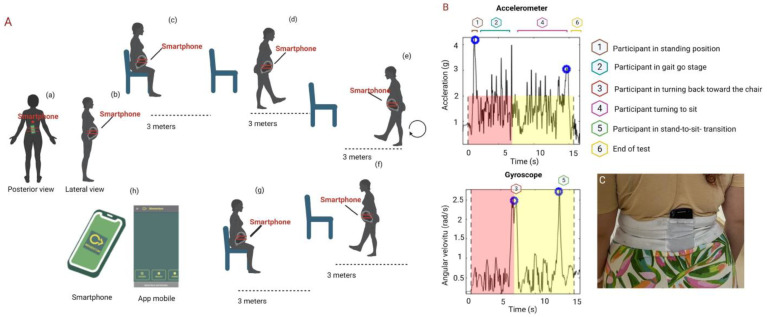
(**A**) representative diagram of the TUG test with gyroscopy and accelerometry is shown. Illustration of the results of the TUG test: posterior view with the smartphone positioned in the lumbar region (a); lateral view with the smartphone positioned in the lumbar region (b); pregnant woman sitting down to start the test (c); transition from sitting to standing (d); moment of the first turn to return (e); moment of return (f); transition from standing to sitting (g); representation of the Momentum app (h). Graphical representation of the gyroscopic and accelerometric data, the red areas stand for the time before the spinning and the yellow areas for the beginning of the spinning (**B**). Positioning the smartphone with the strap (C). This tool was created by “BioRender.com” (2024).

**Figure 5 sensors-24-04615-f005:**
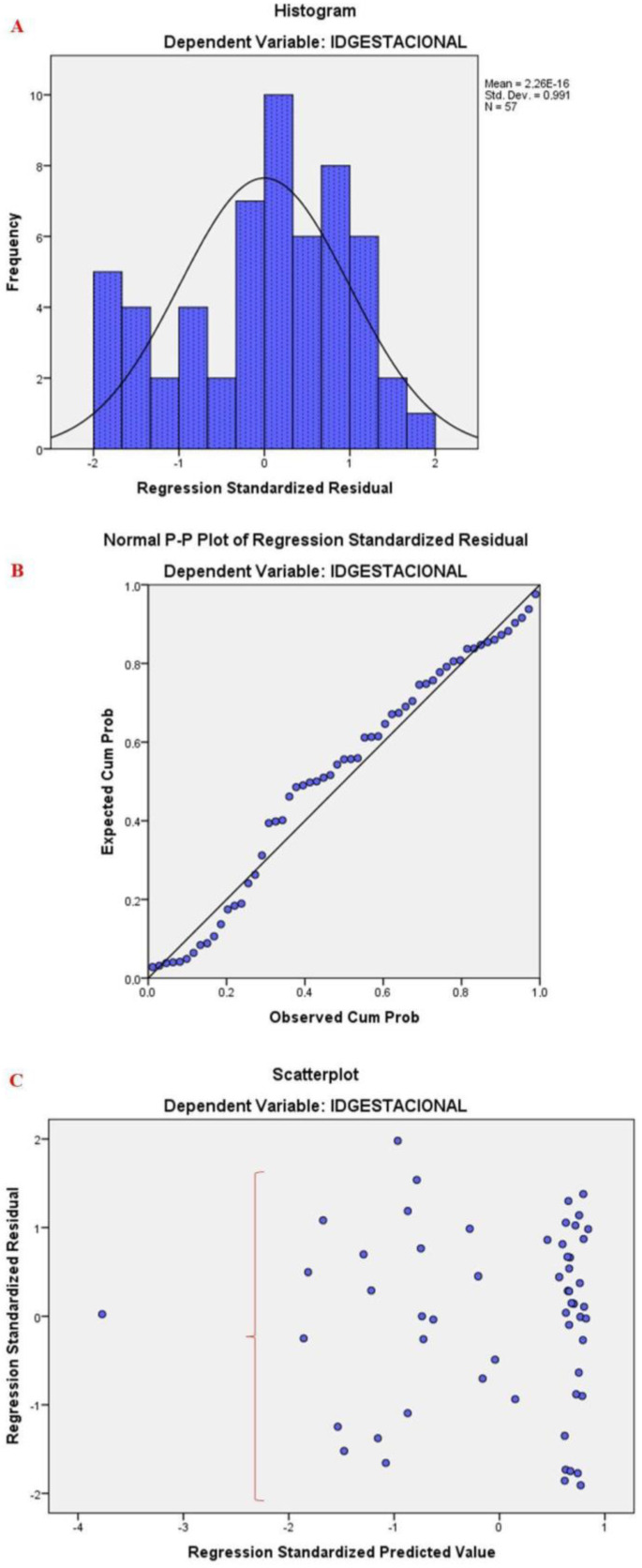
Model assumptions to evaluate the prediction of the myoelectric activity of the pelvic floor muscles, verifying the normality of the residuals (**A**,**B**), and homoscedasticity and linear relationship between the variables (**C**).

**Figure 6 sensors-24-04615-f006:**
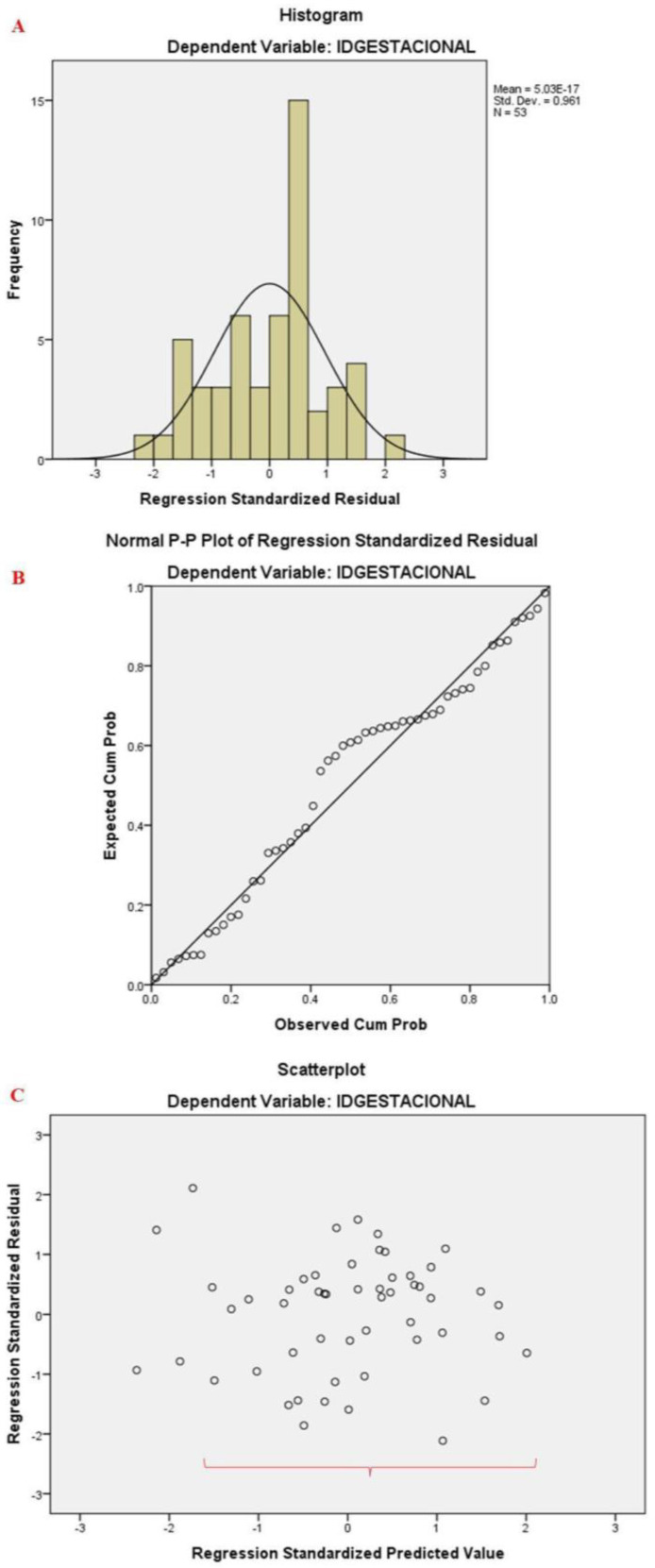
Model assumptions to evaluate the prediction of functional mobility, verifying the normality of the residuals (**A**,**B**), and homoscedasticity and linear relationship between the variables (**C**).

**Figure 7 sensors-24-04615-f007:**
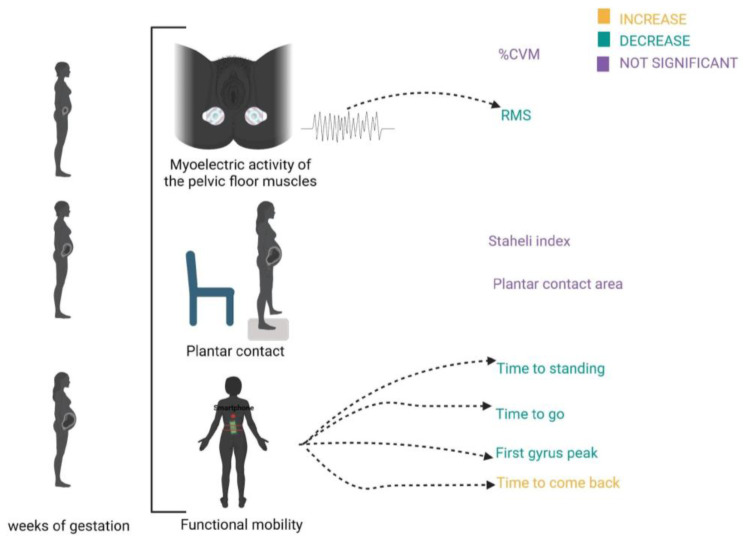
Mapping of the results for the variables analyzed in relation to surface electromyography, plantar contact, and functional mobility.

**Table 1 sensors-24-04615-t001:** Characteristics of the study participants.

	N = 62	%
Race/skin color		
Black	4	6.5
Brown	44	71
White	11	17.7
Yellow	2	3.2
Uninformed	1	1.6
Marital status		
Married	41	66.1
Single	20	32.3
Uninformed	1	1.6
Education (years)		
1 to 4	2	3.2
5 to 9	18	29.1
10 to 12	33	53.2
>12	8	12.9
Uninformed	1	1.6
Family income		
No income	3	4.8
<1	25	40.3
1 to 3	25	40.3
4 to 6	6	9.7
7 to 9	2	3.2
Uninformed	1	1.6

## Data Availability

The data are the responsibility of the authors due to ethical principles and can be made available if requested.

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
