# Peer review of "Influence of Gestational Age on Pelvic Floor Muscle Activity, Plantar Contact, and Functional Mobility in High-Risk Pregnant Women: A Cross-Sectional Study"

_sensors, 2024, doi:10.3390/s24144615_

Round 1
Reviewer 1 Report
Comments and Suggestions for Authors
This manuscript investigates the impact of gestational weeks on myoelectric activity of the pelvic floor muscles, plantar contact area, and functional mobility in high-risk pregnant women. It is well-structured, well-written, concise and thorough. However, I have some minor comments that need to be fixed:
1- The title “The weeks of gestation age influence the myoelectric activity of the pelvic floor muscles, plantar contact and functional mobility in high-risk pregnant women – a cross-sectional study” could be rephrased for clarity. Perhaps the title can be: “Influence of Gestational Age on Pelvic Floor Muscle Activity, Plantar Contact, and Functional Mobility in High-Risk Pregnant Women: A Cross-Sectional Study”.
2- The phrase between line 25 and 28 is not clear: “To investigate the progressive impact of the biomechanical changes that develop during gestational weeks on the myoelectric activity of the pelvic floor muscles and the plantar contact area, as well as to assess the functional mobility of high-risk pregnant women.;”. what the authors are trying to say here? And why there is a semicolon at the end of the phrase ? Please fix.
3- The citations in line 52, are in this format [1;2], while in the discussion section for example in line 407, the authors used another format for citation [45, 46]. The citations should have a consistent format throughout the manuscript. Preferably use this format instead [1,2].
4- The phrase between lines 134 and 137: “Our initial hypothesis was that there was an increase in the EMG of the pelvic floor muscles to increase the lower support of the gravid uterus, increased the plantar contact area and plantar arch collapse to increase the base of support and that there would be a reduction in all parameters during the TUG test.” is not well written, there is no consistency in tense. Please rephrase.
5- It would be better if you include a table that contains all the abbreviations used in the manuscript and their meanings.
Overall, the paper is well-written and provides valuable insights into the biomechanical impacts of pregnancy. Addressing the minor issues mentioned above will enhance the clarity and readability of the manuscript.
Comments on the Quality of English Language
Minor editing of English language required
Author Response
Dear Reviewer,
Please find as an attached file a revised version of our manuscript titled “Influence of Gestational Age on Pelvic Floor Muscle Activity, Plantar Contact, and Functional Mobility in High-Risk Pregnant Women: A Cross-Sectional Study”. All mandatory issues raised have been addressed.
Our responses to the editor’s and reviewers’ comments and corrections to improve the quality of this work can be found below. Please see the highlighted text.
We appreciate the contributions made by the reviewers and hope to have answered all issues accordingly.
Thank you for your attention.
Sincerely yours
REVIEWER 1:
Comments 1: “1- The title “The weeks of gestation age influence the myoelectric activity of the pelvic floor muscles, plantar contact and functional mobility in high-risk pregnant women – a cross-sectional study” could be rephrased for clarity. Perhaps the title can be: “Influence of Gestational Age on Pelvic Floor Muscle Activity, Plantar Contact, and Functional Mobility in High-Risk Pregnant Women: A Cross-Sectional Study”
Response 1: Thank you for your comments and suggestions. We have adopted the suggested title. Please see the highlighted text on Line 2.
Comments 2: “2- The phrase between line 25 and 28 is not clear: “To investigate the progressive impact of the biomechanical changes that develop during gestational weeks on the myoelectric activity of the pelvic floor muscles and the plantar contact area, as well as to assess the functional mobility of high-risk pregnant women.;”. what the authors are trying to say here? And why there is a semicolon at the end of the phrase ? Please fix.”
Response 2: Thank you for your comment. We have made the suggested changes. We have revised the sentence for clarity and removed the semicolons that appeared throughout the summary in the section “Transition”. The revised sentence can be found in the highlighted text on lines 25 to 28.
Comments 3: “3- The citations in line 52, are in this format [1;2], while in the discussion section for example in line 407, the authors used another format for citation [45, 46]. The citations should have a consistent format throughout the manuscript. Preferably use this format instead [1,2].”
Response 3: Thank you for your comment. We have corrected the formatting of the textual citations throughout the manuscript using the suggested preferred format.
Comments 4: “4- The phrase between lines 134 and 137: “Our initial hypothesis was that there was an increase in the EMG of the pelvic floor muscles to increase the lower support of the gravid uterus, increased the plantar contact area and plantar arch collapse to increase the base of support and that there would be a reduction in all parameters during the TUG test.” is not well written, there is no consistency in tense. Please rephrase.”
Response 4: Thank you for your comment. As suggested, the sentence has been rewritten to make the text clearer and more concise. Please see the highlighted text between lines 134 and 138.
Comments 5: “5- It would be better if you include a table that contains all the abbreviations used in the manuscript and their meanings.”
Response 5: Thank you for your comment. However, the abbreviations were made following the Instructions for Authors of "Sensors": "Acronyms/Abbreviations/Initialisms should be defined the first time they appear in each of three sections: the abstract; the main text; the first figure or table. When defined for the first time, the acronym/abbreviation/initialism should be added in parentheses after the written-out form." Available at: https://www.mdpi.com/journal/sensors/instructions
Note: I appreciate your feedback. I would like to inform you that the article has undergone a new English revision by the Curie team (AJE - part of Springer Nature). We apologize for any inconvenience previously caused.

Reviewer 2 Report
Comments and Suggestions for Authors
This manuscript investigates the progressive impact of the biomechanical changes that develop during gestational weeks and assess the functional mobility of high-risk pregnant women. No doubt that this research has some practical significance. But the following issues should be considered to improve the quality of the manuscript carefully.
1. In experiment analyses and conclusions, it seems that pelvic floor myoelectric activity has not helpful association with the increase of gestational age. It is not a good expression for other research because it could not bring big valuable advice. It is a suggestion that further relationship between physiological mechanism and measurement methods should be considered.
2. The most significant part of the manuscript is high-risk estimation of pregnant women. But it seems that floor muscles and the plantar contact area could not give a clear risk evaluation.
3. The manuscript considers inertial sensors in smartphones combined with the TUG could be an assessment tool to monitor. Actually, these sensors are designed for consumer electronics and interfered in many applications by noise. More serious measurements need to be considered.
4. The expression of the manuscript is not satisfied. There are many type errors that should be checked, such as Equation 1. Line Spacing of Table 1 is too large. Some figures should have necessary explanations in the main text. Is Appendix 1 necessary?
5.Could the title of the manuscript be more concise
Comments on the Quality of English Language
The expression should be improved.
Author Response
Dear Reviewer,
Please find as an attached file a revised version of our manuscript titled “Influence of Gestational Age on Pelvic Floor Muscle Activity, Plantar Contact, and Functional Mobility in High-Risk Pregnant Women: A Cross-Sectional Study”. All mandatory issues raised have been addressed.
Our responses to the editor’s and reviewers’ comments and corrections to improve the quality of this work can be found below. Please see the highlighted text.
We appreciate the contributions made by the reviewers and hope to have answered all issues accordingly.
Thank you for your attention.
Sincerely yours
REVIEWER 2:
Comments 1:“1. In experiment analyses and conclusions, it seems that pelvic floor myoelectric activity has not helpful association with the increase of gestational age. It is not a good expression for other research because it could not bring big valuable advice. It is a suggestion that further relationship between physiological mechanism and measurement methods should be considered.”
Response 1: Thank you for your comments. However, as pointed out in lines 350 and 351 and Figure 5, our analyses revealed a reduction in the myoelectric activity (RMS) of the pelvic floor muscles with increasing gestational age. This finding has a direct impact on the need for improvement in clinical practice due to the characteristics of pregnancy. In this way, it is of fundamental importance that professionals rehabilitate these muscles and not just carry out conservative treatment or rest. Therefore, these clinical cases should be discussed by a multiprofessional team. Please see the second and third paragraphs of the discussion, the text of which has been improved to clarify this situation.
Comments 2: “2. The most significant part of the manuscript is high-risk estimation of pregnant women. But it seems that floor muscles and the plantar contact area could not give a clear risk evaluation.”
Response 2: Thank you for your comments. However, it is important to note that the high-risk classification is the target population for our study. This population was chosen for the study because of difficulties during clinical interventions. We did not assess the risk of having high-risk pregnancies using these parameters (floor muscles and the plantar contact area).
Comments 3: “3. The manuscript considers inertial sensors in smartphones combined with the TUG could be an assessment tool to monitor. Actually, these sensors are designed for consumer electronics and interfered in many applications by noise. More serious measurements need to be considered.”
Response 3: Thank you for your comments, and we understand your concern. However, we would like to point out that we used MATLAB to remove interference noise from the smartphone's inertial sensors, as described in the methods section on lines 287 to 288. We believe that this approach significantly improves the accuracy of the measurements, making the sensors a viable tool for monitoring.
Comments 4: “4. The expression of the manuscript is not satisfied. There are many type errors that should be checked, such as Equation 1. Line Spacing of Table 1 is too large. Some figures should have necessary explanations in the main text. Is Appendix 1 necessary?.
Response 4: Thank you for your comments. We have changed the way Equation 1 is presented in the manuscript, as it is described in the text, which can be seen in the highlighted text on line 284. We have corrected the spacing between lines, which can be seen in Table 1. We have removed Appendix 1 from the manuscript, as suggested. We hope that these changes will improve the clarity and presentation of our work.
Comments 5: “5.Could the title of the manuscript be more concise”
Response 5: Thank you for your suggestion. We have adopted your recommendation and changed the title to “Influence of Gestational Age on Pelvic Floor Muscle Activity, Plantar Contact, and Functional Mobility in High-Risk Pregnant Women: A Cross-Sectional Study”, making it more concise and coherent. Please see the highlighted title on Line 2.
Note: I appreciate your feedback. I would like to inform you that the article has undergone a new English revision by the Curie team (AJE - part of Springer Nature). We apologize for any inconvenience previously caused.

Round 2
Reviewer 2 Report
Comments and Suggestions for Authors
All my concerns have been addressed.